# Selection and Evaluation of mRNA and miRNA Reference Genes for Expression Studies (qPCR) in Archived Formalin-Fixed and Paraffin-Embedded (FFPE) Colon Samples of DSS-Induced Colitis Mouse Model

**DOI:** 10.3390/biology12020190

**Published:** 2023-01-26

**Authors:** Ana Unkovič, Emanuela Boštjančič, Aleš Belič, Martina Perše

**Affiliations:** 1Medical Experimental Centre, Institute of Pathology, Faculty of Medicine, University of Ljubljana, 1000 Ljubljana, Slovenia; 2Institute of Pathology, Faculty of Medicine, University of Ljubljana, 1000 Ljubljana, Slovenia; 3Statistics and Modelling, Technical Development Biologics, Novartis Technical Research & Development, Lek Pharmaceuticals d.d., 1000 Ljubljana, Slovenia

**Keywords:** reference gene, archived FFPE, DSS model, mouse, mRNA, miRNA

## Abstract

**Simple Summary:**

Quantitative real-time polymerase chain reaction is a widely used molecular technique in human and animal diagnostics and research. However, its accuracy and reliability are significantly affected by the stability of the selected reference genes. We evaluated the expression stability of reference genes in archived formalin-fixed and paraffin-embedded (FFPE) samples of C57BL/6JOlaHsd mice (males and females) from colitis (DSS-induced) experiments. Our results imply that the FFPE procedure does not change the ranking of stability of reference genes obtained in fresh tissues, which suggests that FFPE samples can be used as a source for further reference gene identification processes. Since miRNA is more stable than mRNA, archived FFPE samples offer an invaluable source for miRNA research (i.e., retrospective analysis, identification of biomarkers, and evaluation of robust reference genes) without any additional use of animals. In addition, multivariate analysis showed that the histological picture is an important factor affecting the expression levels of target genes. Thus, FFPE samples as a source for RNA analyses have an added advantage not only in terms of reduction of animal use, but also in terms of accuracy of results (expression in correlation to the histological picture).

**Abstract:**

The choice of appropriate reference genes is essential for correctly interpreting qPCR data and results. However, the majority of animal studies use a single reference gene without any prior evaluation. Therefore, many qPCR results from rodent studies can be misleading, affecting not only reproducibility but also translatability. In this study, the expression stability of reference genes for mRNA and miRNA in archived FFPE samples of 117 C57BL/6JOlaHsd mice (males and females) from 9 colitis experiments (dextran sulfate sodium; DSS) were evaluated and their expression analysis was performed. In addition, we investigated whether normalization reduced/neutralized the influence of inter/intra-experimental factors which we systematically included in the study. Two statistical algorithms (NormFinder and Bestkeeper) were used to determine the stability of reference genes. Multivariate analysis was made to evaluate the influence of normalization with different reference genes on target gene expression in regard to inter/intra-experimental factors. Results show that archived FFPE samples are a reliable source of RNA and imply that the FFPE procedure does not change the ranking of stability of reference genes obtained in fresh tissues. Multivariate analysis showed that the histological picture is an important factor affecting the expression levels of target genes.

## 1. Introduction

Quantitative real-time polymerase chain reaction (qPCR) is a widely used molecular technique in human and animal diagnostics and research due to its accessibility, sensitivity, specificity, speed, reproducibility, and accuracy [1,2,3,4]. However, qPCR accuracy and reliability are significantly affected by the stability of the selected reference genes. Today, it is clear that universal reference genes do not exist. The expression stability of qPCR reference genes is influenced by many factors, such as species [5], genetic background [4], sex [6], the disease model [7,8,9], and even the disease state (phases of the disease) [10] and organ analyzed [11,12].

A recent systematic review showed that the majority of animal studies (particularly on rodents) use a single reference gene without any prior analysis of reference gene expression stability. Moreover, reference genes identified as the most frequently used single reference gene in rodent studies (such as GAPDH or ACTB) have been shown as inappropriate in numerous experimental settings [13]. For instance, more than 700 published research articles on the dextran sulfate sodium (DSS)-induced colitis model (DSS model) used GAPDH as a single reference gene [7]. Therefore, many qPCR results from past rodent studies can be misleading, affecting not only reproducibility but also translatability. 

Recent advances in methodology have made it possible to extract and analyze RNA or even recover proteins from formalin-fixed and paraffin-embedded (FFPE) tissues [14]. Today, commercially available reagents designed for FFPE tissue and appropriate methodology for RNA analysis are already used in human pathology [15]. In humans, archived FFPE tissues together with associated diagnostic records have been recognised as an invaluable source for molecular research and retrospective analysis [16]. In animal models, pathology is usually an essential part of the study. FFPE samples together with the experimental records are archived for longer periods of time. The archived FFPE samples from animal studies thus represent a great potential for further research without any additional suffering or use of animals. 

To evaluate the appropriateness of archived FFPE samples from animal studies for RNA analyses, we analyzed archived FFPE samples from nine DSS mouse model experiments. The DSS mouse model is one of the most frequently used animal models [17,18]. Recently, a study showed that DSS treatment significantly affects the stability of numerous reference genes [7]. Among the 13 most frequently used reference genes, only a few of them were shown as stably expressed and appropriate. Selection and evaluation has been performed in fresh frozen colon samples. The study involved a very homogenous experimental setting, evaluating only the control and DSS groups, composed of 6 male mice housed in one cage per group and euthanized on day 5 of DSS treatment [7]. The DSS model has many advantages (i.e., not expensive and simple to induce and control). However, there are many factors that can affect the reproducibility of the model (i.e., genetic background, microbiological and environmental factors, DSS treatment) [18], and many of which may also affect the stability of reference genes. To our knowledge, no research has been focused on inter/intra-experimental factors involved in the stability of reference genes in animal models and no study has been performed to identify reference genes for small RNA in the DSS model, although microRNA (miRNA)-related research is currently one of the fastest-growing topics. In addition, no animal study evaluated the stability of reference genes (mRNA/miRNA) in archived FFPE samples.

Therefore, the first aim of our study was to evaluate the stability of selected reference genes (mRNA/miRNA) in archived FFPE colon samples from nine DSS experiments. The purpose was to select the best reference genes from those that have been already suggested by others for fresh frozen tissue based on screening data. In the case of mRNAs, the purpose was to check whether the reference genes were also suitable for the FFPE tissues. In the case of miRNAs (since there are no data on reference gene selection in the DSS model) the purpose was to test the expression stability of those miRNAs that are suggested by the manufacturer as most stably expressed among human and mouse tissues. The second aim was to select and evaluate the most appropriate reference genes (mRNA/miRNA) for FFPE samples that would not be affected by DSS treatment and inter/intra-experimental variability. The third aim was to demonstrate how the selection of reference genes affects qPCR data normalization and results. The influence of the reference gene selection on qPCR data normalization was evaluated by multivariate analysis as well.

## 2. Materials and Methods

### 2.1. Animals and Experimental Protocol

Seven-to-twelve-week-old C57BL/6JOlaHsd mice (Harlan, Italy or Medical Experimental Centre, Ljubljana, Slovenia) were housed in groups of 5–8 mice per cage (Ehret, Germany; 825 cm^2^ floor area) with bedding (Lignocel 3/4, Germany) and cellulose towels. Mice were maintained in controlled environmental conditions with a 12-h light/dark cycle (7 am/7 pm) and ad libitum access to tap water and rodent diets (Altromin 1324, Germany or Teklad 2018, Italy). Colitis was induced by the addition of DSS (2–3% DSS, TdB Consultancy AB, Uppsala Sweden, molecular weight 40 kDa) to drinking water for 4–6 days. The DSS solution was freshly made every morning. After the period of DSS treatment, mice received drinking water until euthanized with CO^2^ (0 days to 3 months) (Appendix A). At the autopsy, all internal organs except the central nervous system were examined. The whole colon was removed, opened wide, formalin fixed and stained with the Kreyberg-Jareg method. FFPE samples were stored at a room temperature in the archives of the Medical experimental centre. Animal experiments were performed in accordance with Directive 2010/63/EU and Slovenian legislation (approved by the National Ethical Committee a permit no. 34401-54/2012/5 and U34401-11/2017/4 from the Food Safety, Veterinary and Plant Protection Administration). The study was reported following the ARRIVE guidelines.

### 2.2. Archived FFPE Colon Samples from 9 DSS Experiments and Selection

Our study was performed on archived FFPE colon samples of DSS-treated C57BL/6JOlaHsd mice involved in 9 DSS experiments performed at the Medical Experimental Centre (Ljubljana, Slovenia) from December 2010 to July 2014 (Appendix A). FFPE samples from the control (untreated) and DSS-treated mice were included in the study according to the following criteria:FFPE samples from 9 DSS experiments;FFPE samples from females and males;FFPE samples from the distal part of the colon (last 1/3);FFPE samples have been histological examined and confirmed that:
a.all layers of the colon wall and no other tissue is present on the FFPE samples;b.all samples from the DSS group have histologically confirmed colitis;c.all samples from the control group show a normal histological picture (no signs of lesion or inflammation).

Among selected FFPE samples, computer randomization was performed to get a representative number of both sexes from each DSS experiment. Finally, the following number of samples was obtained:in the control group: 20 FFPE samples (from 20 untreated healthy mice: 10 males and 10 females),in the DSS group: 97 FFPE samples (from 97 DSS-treated mice: 50 males and 47 females).

Altogether, 117 FFPE samples from 9 DSS experiments were included in the analysis. Among the experiments, there were similarities in the experimental settings such as strain (C57BL/6JOlaHsd), age (7–12 weeks), origin (Harlan or MEC), and differences such as sex (female, male), DSS treatment (concentration of DSS (2–3%), duration of DSS treatment (4–6 days), duration of inflammation (0 days–3 months), environmental factors (time of experiment execution (2010–2014), rodent diet (Altromin, Teklad, batch), histological picture (mild-severe inflammation, mucosal/transmural, erosion), and FFPE sample factors such as fixation time (1 week–3 weeks) and duration of storage. Storage conditions were the same for all paraffin blocks. In our laboratory, autopsy, sampling, and formalin fixation were done under the same environmental conditions, using the same necropsy protocol.

### 2.3. RNA Isolation from Archived FFPE Samples

Total RNA was isolated using a commercially available miRNeasy FFPE Kit (Qiagen, Hilden, Germany), specially designed for purifying total RNA from FFPE tissues. Isolation was performed according to the manufacturer’s instructions from four 10 μm thick paraffin slices from each paraffin block. Isolation was based on binding nucleic acids to column membranes. The miRNeasy FFPE Kit provided an adapted method to overcome formalin crosslinking and enabled the efficient release of RNA without compromising integrity. It also enabled elution volumes from 10 to 30 μL, which is important in cases of limited sample quantity. All reagents, besides ethanol (Sigma-Aldrich and Merck, Kenilworth, NJ, USA), were supplied by the manufacturer Qiagen. The quantity and purity of isolated RNA were assessed spectrophotometrically by measuring absorbance at 260 nm and detecting impurities at 230 and 280 nm, respectively, using NanoDrop One (Thermo Fisher Scientific, Waltham, MA, USA). All our samples had an absorption ratio of A260/280 at 1.8 or higher, indicating that isolated RNA from FFPE samples was of good purity [19,20]. 

### 2.4. Selection of Reference Genes (mRNA/miRNA) for Archived FFPE Samples

The selection of mRNA candidate reference genes was based on the study evaluating the stability of the 13 most commonly used reference genes in DSS-treated mice isolated from fresh frozen colon tissue [7]. Only candidate reference genes that showed quantification cycle (Cq)-values lower than 30 and the highest stability among 13 candidate reference genes evaluated on fresh colon samples previously by Eissa et al. (i.e., EEF2, RPLP0, TBP, NONO, PPIA) [7] were selected. Candidate reference genes for miRNAs (miR-191-5p, miR-103a-3p, and miR-16-5p) were selected based on previous results from miRNA expression analysis in human IBD [21,22], while small nuclear RNA (U6) is commonly used in the DSS mouse model [23,24,25,26,27]. After the stability determination of candidate reference genes, evaluation of the most appropriate mRNA reference genes was assessed using target gene expression analysis of two well-known molecular factors involved in colitis (TNFR1 and TNFR2) [28,29,30,31,32,33,34]. An evaluation was also made for the most appropriate miRNA reference genes. Target miRNAs were selected based on their elevated expression in colitis [23,35]. We tested the expression efficiency of seven target miRNAs (miR-223-3p, miR-181a-5p, miR-680, miR-1224-5p, miR-5128, miR-3968, and let-7f-5p). Only two miRNAs (miR-223-3p and miR-181a-5p) had an appropriate amplification efficiency value and were selected for further evaluation. Hydrolysis detection probes (TaqMan^®^ Gene Expression Assay, Applied Biosystems, Foster City, CA, USA) used are listed in Table 1, and miRCURY LNA miRNA PCR Assays (Qiagen, Hilden, Germany) in Table 2.

### 2.5. Reverse Transcription and Quantitative Real-time Polymerase Chain Reaction (RT-qPCR)

#### 2.5.1. mRNA

RT-qPCR reactions were performed within a 384wellplate in a QuantStudio™ 7 Pro (Applied Biosystems, Waltham, MA, USA) using a Luna^®^ Universal One-Step RT-qPCR Kit (New England Biolabs, Inc., Ipswich, MA, USA) and hydrolysis probes (TaqMan^®^ detection probes, Applied Biosystems, Foster City, CA, USA) (Table 1). The Luna^®^ Universal One-Step RT-qPCR Kit enabled quick setup and limited handling (less pipetting; reduction of potential errors), and was more time-efficient than two-step commercially available kits. It also enabled high amounts of sample input, which is relevant for applications where RNA is present in low abundance. In each individual reaction, there was 10 ng of RNA and a final volume of 10 µL. The RT and PCR protocol were as follows: 55 °C for 10 min, 95 °C for 1 min, followed by 45 cycles at 95 °C for 15 s and at 60 °C for 1 min. Positive and negative controls (with no RNA template) were used for each hydrolysis probe. All RT-qPCRs were run in duplicates.

#### 2.5.2. miRNA

RT reactions were performed in a PTC-200 Thermal Cycler (MJ Research, Waltham, MA, USA) using a miRCURY LNA RT Kit (Qiagen, Hilden, Germany), according to the manufacturer’s instructions. In each individual reaction was 10 ng of RNA. The RT protocol was as follows: 60 min at 42 °C and 5 min at 95 °C. cDNAs were stored at −20 °C until qPCR. qPCR reactions were performed within a 384-well plate in a QuantStudio™ 7 Pro (Applied Biosystems, Waltham, MA, USA) using a miRCURY LNA SYBR^®^ PCR Kit (Qiagen, Hilden, Germany) and miRCURY LNA miRNA PCR Assays (Qiagen, Hilden, Germany) (Table 2), in reactions with a final volume of 10 µL. In each individual reaction was 3 µL of 1:60 diluted cDNA. The qPCR protocol was as follows: 2 min at 95 °C, followed by 40 cycles at 95 °C for 10 s, and at 56 °C for 1 min, followed by melting curve analysis at 60–95°C. Positive and negative controls (with no RNA template) were used for each miRNA PCR Assay. All qPCRs were run in duplicates.

### 2.6. Amplification Efficiency and RNA Integrity

Amplification efficiencies in the exponential phase were calculated for each individual mRNA/miRNA in each group of samples (control and DSS group) using standard curves (5-point serial dilution of pooled RNA containing equal amounts of samples, 4-fold serial dilution for miRNA, and 5-fold serial dilution for mRNA). All qPCR reactions for amplification efficiency testing were performed in triplicate. The linear dynamic range for each gene was determined according to standard curves and correlation coefficients (R2).

As the amplicon of one of the targeted reference genes was 112 bp long, its successful amplification can be used as an indication that isolated RNA from FFPE samples is of acceptable quality for further analysis.

**Table 1 biology-12-00190-t001:** Information on mRNA candidate reference genes and target genes and their qPCR amplification characteristics in archived FFPE colon samples of control and DSS-treated C57BL6/JOlaHsd mice. Legend: Eff., amplification efficiency; R2, correlation coefficient.

					Control Group		DSS Group	
Gene Symbol	Gene Name	Assay ID	Length [bp]	Catalogue No.	Eff. [%]	R^2^	Eff. [%]	R^2^
EEF2	Eucaryotic translation elongation factor 2	Mm01171434_g1	74	4331182	95	0.998	102	0.997
TBP	TATA box binding protein	Mm00446971_m1	93	4331182	125	0.974	107	0.971
NONO	Non-POU domain containing, octamer binding protein	Mm07293722_g1	77	4351372	83	0.993	103	0.990
PPIA	Peptidylprolyl isomerase A	Mm02342429_g1	112	4331182	90	0.998	85	0.998
RPLP0	Ribosomal protein large P0	Mm01974474_gH	89	4331182	89	0.998	91	0.997
TNFR1	Tumor necrosis factor receptor 1	Mm00441875_m1	69	4331182	98	0.992	99	0.990
TNFR2	Tumor necrosis factor receptor 2	Mm00441889_m1	64	4331182	98	0.994	92	0.994

**Table 2 biology-12-00190-t002:** Information on miRNA candidate reference genes and target genes and their qPCR amplification characteristics in archived FFPE colon samples of control and DSS-treated C57BL6/JOlaHsd mice. Legend: Eff., amplification efficiency; R2, correlation coefficient.

			Control Group		DSS Group	
Mirna Symbol	Accession Number	Gene Globe ID/Cat. No.	Eff. [%]	R^2^	Eff. [%]	R^2^
U6 snRNA	/	YP02119464/339306	106	0.995	104	0.994
miR-191-5p	MIMAT0000440	YP00204306/339306	109	0.995	107	0.995
miR-103a-3p	MIMAT0000101	YP00204063/339306	106	0.995	103	0.996
miR-16-5p	MIMAT0000069	YP00205702/339306	100	0.997	96	0.998
miR-181a-5p	MIMAT0000256	YP00206081/339306	102	0.990	105	0.997
miR-223-3p	MIMAT0000280	YP00205986/339306	101	0.994	93	0.998
miR-680	MIMAT0003457	YP00205101/339306	121	0.990	92	0.842
miR-1224-5p	MIMAT0005460	YP02115039/339306	151	0.885	145	0.882
miR-5128	MIMAT0020639	YP02111071/339306	119	0.613	138	0.082
miR-3968	MIMAT0019352	YP02104299/339306	112	0.933	82	0.911
Let-7f-5p	MIMAT0000067	YP00204359/339306	72	0.996	70	0.996

### 2.7. Reference Gene Stability Analysis

Reference gene stability in archived FFPE samples was evaluated using publicly-available tools based on validated assessment algorithms, i.e., NormFinder [36] and BestKeeper [37].

BestKeeper identifies the gene expression stability of candidate reference genes based on Cq-values. The algorithm calculated the standard deviation (SD) of the Cq-values. The reference gene having the lowest SD-value was considered the most stably expressed [37].

NormFinder identifies the optimal normalization gene (most stably expressed gene), stability values of each candidate reference gene, the best combination of two genes for a two-gene normalization factor, and their stability value. Reference genes with the lowest stability value were considered the most stable. Normfinder also identifies intra- and intergroup variations based on Cq-values [36].

### 2.8. Statistical Analysis

For statistical analysis of the expression of reference genes, Cq-values were used. For statistical analysis of the expression of target genes, ∆Cq-values were used, and for summarizing influence and graphical presentation of used reference genes on the expression of selected target genes, fold change was used. Both calculations were based on the publication by Latham et al, 2010 [38]. The results were compared and analyzed using IBM SPSS Statistics, version 247 (SPSS Inc., Chicago, IL, USA) and the Mann–Whitney U test (for independent groups of samples). In all statistical tests, the differences were marked as statistically significant at a value of *p* ≤ 0.05. Excel 2016 (Microsoft Corporation, Washington, DC, USA) was used to graphically display the results.

### 2.9. Multivariate Analysis

Two methods were used for multivariate data analysis. 

Principal component analysis (PCA) [39] was used to evaluate the efficiency of the computer randomization of samples. This is important for the later evaluation of multivariate model relevance. If data are not sufficiently randomized in model inputs, the resulting models may have limited prediction value. The PCA algorithm description is available in Appendix A. PCA transforms the original set of variables describing a studied system into an uncorrelated set of principal components. The principal components represent a more natural set of variables that describe the studied system. PCA also provides information on how much of the total data set variability is covered by a specific principal component, allowing for a reduction of the original data set with minimal information loss. Principal components contributing the least to the total variability and that cumulatively contribute less than the estimated uncertainty of the measurement methods can be omitted from the data set. The data set for modeling is divided into model inputs (covariates, independent variables, x-variables) and model outputs (dependent variables, y-variables). If many principal components can be omitted when analysing model inputs, this means that the original set of variables is highly correlated and, therefore, the multivariate model will have limited validity.

Partial least squares (PLS) [40,41] is a multivariate modeling method in which PCA is used to decorrelate model input and output variables, and then multiple regression is applied on decorrelated variables. The method is iterative, so the original PCA decomposition is adapted to provide the best possible model fit. When model input variables are not correlated, PLS provides the same model as multiple regression. Model loadings relate model inputs to model outputs and can be used to order the input variables according to the level of influence they have on the output variables. Commonly, the PLS model is validated on the basis of R2, Q2, and permutation tests. R2 is a coefficient of determination and explains how much of the total variability can be explained with the model based on the part of the data set used to estimate the model. Q2 is also a coefficient of determination; however, it is calculated on the part of the data set not used to estimate the model. Q2 is generally smaller than R2, but the difference is not large for good data sets if a model can be obtained. The permutation test mixes input-output data pairs and calculates R2 and Q2 for the model on the mixed data set. If R2 and Q2 for the correct input-output data pairs are higher than for the mixed data set, there is some meaningful relation in the data and the model was able to extract it.

Multivariate data analysis was performed in SIMCA 16 (Sartorius Stedim Data Analytics AB, Umeå, Sweden).

## 3. Results

### 3.1. Histological Examination of Archived FFPE Samples

Prior to RNA isolation from four sections of each FFPE sample, all FFPE samples were histologically examined to meet the criteria described in the method section. FFPE samples from DSS groups showed colitis with erosion and/or various intensity and degree of inflammation. The intensity of inflammation ranged from mild to severe. Inflammation was limited to mucosa, or extending into the submucosa and/or muscular layer, or transmural. FFPE samples from the control group showed a histological picture with no signs of injury or inflammation (Figure 1a).

### 3.2. Efficiency and Specificity of mRNA/miRNA qPCR

The performance of each TaqMan^®^ assay (Table 1) and each miRCURY LNA miRNA PCR assay (Table 2) was determined by qPCR specificity and/or efficiency. To determine the qPCR efficiency, standard curves were made. RNA pools of control and DSS group samples were used as templates. The amplification efficiencies of all miRNA assays were in the range of successful amplification (i.e., 90 and 110%) [19], but not all mRNA assays, as they were between 83 and 125%. To detect a target in a reasonably true linear fashion, R2 should be above 0.98 [42]. All R2-values, except for TBP (R2 =0.97), were greater than 0.99 (Table 1, Table 2, and Appendix A), indicating the accuracy of amplification efficiency. All negative controls (with no RNA/cDNA template) showed no amplification; their Cq-values were undetermined. Specificity was determined as high for all miRCURY LNA miRNA PCR assays, due to a single peak in the melting curve analysis.

### 3.3. Candidate Reference Gene Expression Levels in Archived FFPE Samples

Candidate reference gene expression levels were obtained using Cq-values and statistics in all tested samples.

The mean Cq-values for all candidate mRNA reference genes ranged from 26 to 35. TBP (33.67) had the highest median value and was undetermined in 16 of 117 FFPE samples (all males). EEF2 (26.28) had the lowest median value, indicating its highest abundance among all candidate reference genes in archived FFPE samples. PPIA (with the longest amplicon of 112 bp) was successfully amplified in all FFPE samples, indicating that the isolated RNA was of acceptable quality. All five candidate reference genes had variable expression levels (SD); the lowest was found for EEF2 (SD = 1.748) and the highest for RPLP0 (SD = 2.409) (Table 3 and Figure 2a).

In the case of candidate reference genes for miRNA, the mean Cq-values ranged from 26 to 29, indicating a more similar abundance of all candidate reference genes for miRNA in FFPE samples in comparison with reference genes for mRNA. miR-191-5p (SD = 0.849) showed the lowest variability and U6 (SD = 1.238) showed the highest (Table 3 and Figure 2e).

### 3.4. Experimental Factors Affecting Reference Gene Expression Patterns

An ideal reference gene should be consistently expressed at the same levels in both sexes and under all experimental conditions [43]. In order to select the most suitable candidate reference genes for archived FFPE samples of the DSS mouse model, we examined the effects of DSS treatment and other systematically included inter/intra-experimental factors on their expression profiles (Figure 2). 

#### 3.4.1. Effects of DSS Treatment

To evaluate whether DSS treatment (various phases and gravity of colitis) had any effect on the expression of reference genes, all FFPE samples of the control groups were compared with the DSS groups (Figure 2b,f and Appendix A). 

Results showed that 3/5 candidate reference genes for mRNAs (i.e., EEF2 (*p* = 0.003), PPIA (*p* = 0.018), and NONO (*p* = 0.000)) were significantly affected by DSS treatment, while TBP and RPLP0 were not (Figure 2b). Of all candidate reference genes for miRNA, only miR-103a-3p (*p* = 0.000) was significantly altered by DSS treatment (Figure 2f).

#### 3.4.2. Effects of Sex

Expression patterns of reference genes in females and males were compared both with regard to, and regardless of, DSS treatment (Figure 2c,d,g,h and Appendix A). Results showed that Cq-values and descriptive statistics differed between females and males. Males showed higher SD-values in all tested parameters except U6, indicating lower expression stability of reference genes in males in comparison with females. In the case of U6, higher SD-values were found in females.

The main difference in mRNA was observed in reference gene PPIA, which showed a significant difference in mRNA expression between control and DSS groups in females (*p* = 0.001), but not in males (*p* = 0.226). A minor difference between females (*p* = 0.001) and males (*p* = 0.031) was found also in EEF2 expression. 

Among all miRNA reference genes, only miR-103a-3p showed a significant difference in expression pattern between the control and DSS groups in both sexes, but the difference was higher in males (*p* = 0.001) than in females (*p* = 0.025).

#### 3.4.3. Effects of Inter/Intra-Experimental Factors

To evaluate the effects of inter/intra-experimental factors, the expression patterns of candidate reference genes from nine DSS experiments were compared (Figure 1b). Experiments performed during a four-year period were numbered (E1–E9). Experiments differed in time of execution, and consequently in the duration of FFPE block storage, sex, DSS treatment (concentration and duration of DSS treatment), duration of inflammation (0 days–3 months), and histological picture (Figure 1a,b and Appendix A. The time of formalin fixation ranged from 1–3 weeks. All the above-stated differences were taken into consideration when expression patterns of reference genes were studied. Since FFPE blocks were stored together, the effects of storage conditions was excluded.

We noticed different expression patterns of mRNA reference genes among the experiments. Experiments numbered from E1 to E4 represented only females which showed a quite stable expression of mRNA candidate reference genes, except in the case of NONO, whose Cq-values were altered by DSS treatment. Experiment E5, the only case with females (n = 9) and males (n = 6), showed stable expression of reference genes, regardless of sex. Experiments from E6 to E9 represented only males which showed instability of their Cq-values. We could not find a correlation between the storage duration of FFPE blocks and the expression pattern of reference genes. Reference gene expression stability was not related to the histological picture either, as in the E5 series with stably expressed reference genes, there were histologically heterogeneous samples (from mild to severe inflammation, limited to mucosa/transmural, erosion). 

The expression pattern of reference genes for miRNA among experiments was not as different as it was in the case of mRNA. Experiments from E1 to E9 showed comparable expression patterns of miRNA candidate reference genes. Only in the case of U6, Cq-values were mostly altered in females (E2 and E4) in comparison with males.

**Figure 1 biology-12-00190-f001:**
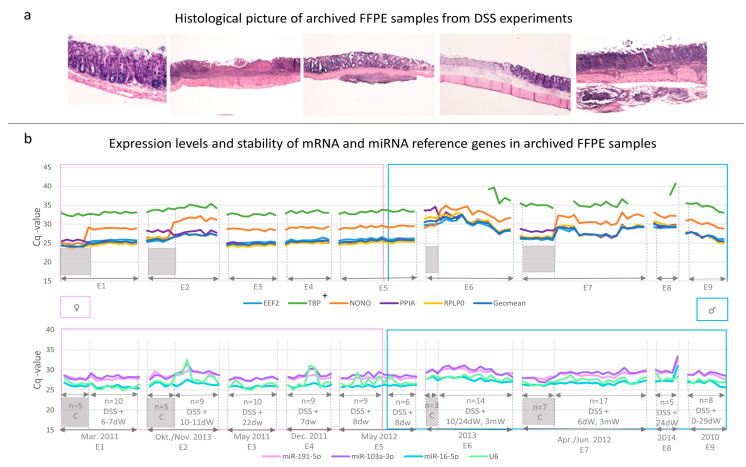
Representative histological pictures and expression levels of candidate reference genes in archived FFPE colon samples of control and DSS treated C57BL6/JOlaHsd mice. (**a**) Representative pictures of archived FFPE colon samples (Kreyberg-Jareg staining) show colon wall histologically evaluated as normal or altered (mild/moderate/severe inflammation, erosion, inflammation limited to the mucosa or extending into deeper layers of colon wall). (**b**) Expression and stability of all candidate reference genes in archived FFPE colon samples from 9 DSS experiments (E1–E9) conducted in year from 2010 to 2014 (n = 117). Graph shows Cq-values of reference genes (y-axis) in all archived FFPE samples shown as (x-axis) number of FFPE samples (n) selected from each experiment (E1–E9), together with the information about DSS treatment (i.e., DSS and control groups separated by sex: females (pink frame) and males (blue frame)) and duration of inflammation (dW, mW). Legend: E1–E9, experiment number; C, control group; DSS, dextran sodium sulphate treatment; dW, days of water; mW, months of water; n, number FFPE samples represent the number of animals; TBP+, 16 of 117 FFPE samples were undetermined for TBP, 2 from the control and 14 from the DSS group, all undetermined FFPE samples were males.

**Figure 2 biology-12-00190-f002:**
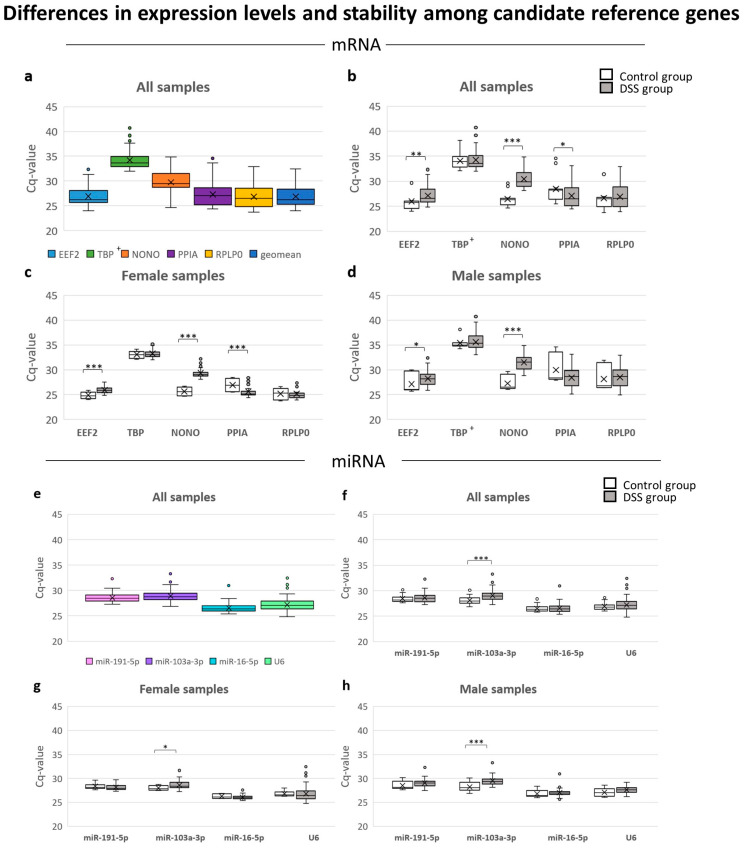
Expression levels and stability of candidate reference genes in archived FFPE colon samples of control and DSS-treated C57BL6/JOlaHsd mice (n = 117). (**a**) mRNA candidate reference genes in FFPE samples regardless of sex or DSS treatment, (**b**) in regard to DSS treatment and (**c**,**d**) sex. (**e**) miRNA candidate reference genes in FFPE samples regardless of sex or DSS treatment, (**f**) in regard to DSS treatment, and (**g**,**h**) sex. Archived FFPE colon samples were selected from 9 DSS experiments conducted during a 4-year period (2010–2014) and were composed of control (n = 20; 10 female, 10 males) and DSS group (n = 97; 47 females, 50 males). Legend: the box indicates the 25th to 75th percentiles; x denotes the mean; the error bars denote the maximum and minimum value; statistical significance: * *p* ≤ 0.05, ** *p* ≤ 0.01, *** *p* ≤ 0.001; TBP†, 16 of 117 FFPE samples were undetermined for TBP, 2 from the control and 14 from the DSS group, all undetermined FFPE samples were males.

### 3.5. Stability of Reference Genes Was Evaluated by BestKeeper and NormFinder

#### 3.5.1. BestKeeper

BestKeeper determines the expression stability value of candidate reference genes based on SD-value. The candidate reference gene with the lowest SD-value has the most stable gene expression. Any candidate reference gene studied with an SD-value higher than 1 is considered inconsistently expressed [37]. SD-values (Table 4) were first calculated separately for the control and DSS groups. It was found that all candidate reference genes for mRNA showed low expression stability (SD-values > 1) in both groups (control and DSS). To investigate potential sex-specific differences in the expression stability of reference genes in DSS colitis, the stability of candidate reference genes was tested between males and females. The results showed that males are more variable in terms of expression stability (higher SD-values) than females. Considering all the results obtained with BestKeeper, EEF2 showed the best expression stability among tested groups regardless of DSS treatment (DSS vs. control), sex, and inter/intra-experimental factors. Although TBP showed low SD-values in many of the tested groups, TBP was undetermined in 16 of 60 males and thus can not be rated as a stably expressed candidate reference gene for mRNA. 

Almost all reference genes for miRNAs showed high stability (SD-values <1), while U6 was found to be unstably expressed in DSS-treated females (Table 4, Figure 1). According to SD-values, miR-16-5p was the most stably expressed reference gene for miRNA.

#### 3.5.2. NormFinder:

The expression stability values of candidate reference genes, the best combination of two reference genes, and their stability values, calculated using NormFinder, are shown in Figure 3, and their intra- and intergroup variation in Appendix A.

The most stable reference gene for mRNA was EEF2, followed by RPLP0. TBP has medium range stability, but 16 of 60 males were undetermined for TBP, and those were not included in the calculations. PPIA and NONO were the least stable reference genes. Excluding TBP Cq-values from the calculations, the stability values of the reference genes were slightly higher; however, the best reference gene and the best combination of two genes remained the same.

The most stable miRNA reference gene was miR-191-5p, followed by U6. Nevertheless, miR-16-5p and miR-103a-3p presented the best combination of two miRNA reference genes, with the lowest stability value (Figure 3).

### 3.6. Multivariate Analysis

First, we evaluated the randomization of samples using PCA. Ten principal components were required to explain 96% of total factor variability, and fourteen components to explain 99.9% of total factor variability, while a total of 20 factors/factor levels were present in the data set. Ideally, with well-designed experiments, all 20 components should be needed to explain that much of the total variability, each component contributing an equal share. While the randomness of the data set was not as high as that of data from formally-designed experiments, it was still high enough to reasonably expect relevant PLS models. To evaluate the influence of normalization with different reference genes on target gene expression in regard to known inter/intra-experimental factors, PLS models were used. The PLS models were designed to predict absolute or relative expression levels (model output) of target genes based on the following inter/intra-experimental factors (model inputs): experiments (E1–E9), sex (female, male), DSS treatment (concentration of DSS, duration of DSS administration), duration of inflammation (0 days–3 months), and histological picture. Contributions of inter/intra-experimental factors were studied through so-called PLS model loadings. PLS model loading describes the relative contribution (influence) of a particular inter/intra-experimental factor to PLS model prediction (expression levels of a gene), where the mean value of the loading (particular inter/intra-experimental factor) as well as its variability are taken into account. Highly variable loadings (with variability higher than the mean value; SD > mean) are regarded as non-significant, indicating that the associate inter/intra-experimental factor does not contribute significantly to the PLS model prediction (i.e., has no significant influence on the expression levels of a gene, in our case). The reported loadings are normalized such that the sum of the squared mean values of the loadings is equal to one. The loadings therefore directly indicate the percentage each intra/inter-experimental factor contributes to the expression level prediction with the PLS model. Figure 4c is a schematic presentation of the influence (relative contribution) of a particular inter/intra-experimental factor on expression levels of TNFR1 (absolute and relative) according to PLS models (columns representing loadings of factors). All PLS models show good prediction statistics. The model for predicting absolute expression levels of TNFR1 (no normalization) has an R2 of 0.86 and a Q2 of 0.83; the model for relative expression of TNFR1 normalized to EEF2/RPLP0 has an R2 of 0.65 and a Q2 of 0.57; the model for relative expression of TNFR1 normalized to PPIA has an R2 of 0.76 and a Q2 of 0.6; and the model for relative expression of TNFR1 normalized to NONO has an R2 of 0.68 and a Q2 of 0.61. An R2 value closer to 1 indicates that, in the PLS models, the majority of factors that can affect the expression level are included. Permutation tests also show high model prediction power.

In addition, Figure 4 demonstrates how the selection of reference genes can drastically affect the relative contribution of a particular inter/intra-experimental factor on the relative expression of a target gene. Different normalizations may result in inverted correlations of target genes to inter/intra-experimental factors. For instance, while sex (female) is positively correlated with TNFR1 expression pattern if TNFR1 expression is not normalized, or is normalized to NONO, it is negatively correlated to TNFR1 expression if it is normalized to the combination of EEF2/RPLP0 or PPIA. On the other hand, Figure 4 demonstrates that the type of histological picture (mild/moderate/severe inflammation; erosion; heterogenous) affects the expression levels of TNFR1 (histological picture correlation is affected), and normalization can completely change the magnitude of the effect that each type of histological picture contributes to TNFR1 expression. Normalization also has a significant effect on experiment-specific (E1–E9) correlations to TNFR1 expression.

PLS models for the TNFR2 expression pattern, with and without normalizations, were also constructed. Similarly to the PLS models for TNFR1, they all show favourable statistics in terms of predictability and sensitivity to inter/intra-experimental factors (Appendix A). The model for predicting absolute expression levels has an R2 of 0.77 and a Q2 of 0.72; the model for relative expression of TNFR2 normalized to EEF2/RPLP0 has an R2 of 0.71 and a Q2 of 0.64; the model for relative expression of TNFR2 normalized to PPIA has an R2 of 0.61 and a Q2 of 0.51; and the model for relative expression of TNFR2 normalized to NONO has an R2 of 0.64 and a Q2 of 0.56. Permutation tests show high sensitivity to factors.

Similar findings to those observed in TNFR1 expression patterns can also be seen in the case of TNFR2. A reference gene, or a combination of reference genes, has a significant effect on how particular inter/intra-experimental factor affect the TNFR2 expression.

Multivariate analysis was performed for miRNA as well (Appendix A). The PLS models for miR-181a-5p and miR-223-3p showed favourable statistical properties for prediction and sensitivity to inter/intra-experimental factors when predicting absolute expression levels (R2 of 0.59 and Q2 of 0.53) and relative expression of miR-181a-5p vs. miR-191-5p/miR-103a-3p/miR-16-5p (R2 of 0.57 and Q2 of 0.44); however, no significant PLS model could be obtained for prediction of relative expression of miR-181a-5p normalized to U6 (R2 of 0.1 and Q2 of -0.07, failed permutation test).

The PLS model for miR-223-3p showed relative expression of miR-223-3p normalized to miR-191-5p/miR-103a-3p/miR-16-5p (R2 of 0.57 and Q2 of 0.44); however, no significant PLS model could be obtained for prediction of relative expression of miR-223-3p normalized to U6 (R2 of 0.19 and Q2 of 0.08).

PLS models demonstrate that normalization generally decreased the correlation of target gene expression with experiment number (E1–E9), indicating that the normalization of the target gene reduces non-specific inter/intra-experiment differences; however, the results also show that there are still other factors that can influence target gene expression levels (remain significant regarding the PLS model loadings).

### 3.7. Influence of the Use of Different Reference Genes on Expression Analysis of Selected Target Genes

Based on comparative analysis of males and females from control and DSS groups obtained from nine experiments (inter/intra-experimental factors), and with the help of different approaches (BestKeeper, NormFinder and multivariate analysis), we propose that, among the selected genes, EEF2 and RPLP0 (EEF2/RPLP0) are the most suitable combination of mRNA reference genes, and miR-191-5p, miR-103-3p, and miR-16-5p (miR-191-5p/miR-103-3p/miR-16-5p) are the best combinations of reference genes for miRNAs for reliable normalization of qPCR results in archived FFPE samples of DSS-treated C57BL/6JOlaHsd mice of both sexes.

To study the effects of selected reference genes on target gene expression, the expression of TNFR1 and TNFR2 was normalized to the most stable reference gene combination (EEF2/RPLP0) and to the least stable reference genes (PPIA and NONO) (Figure 4a). Normalization of target gene expression was made in groups of samples that differ in factors that have been shown to significantly affect the expression stability of reference genes (i.e., DSS treatment and sex). When TNFR1 and TNFR2 were normalized to the most stable reference genes (EEF2/RPLP0), both target genes showed upregulation in the DSS group in comparison with control groups, while normalization of the least stable reference gene, PPIA, to one showed downregulation of TNFR1 and lower folds of TNFR2. In contrast, normalization to NONO showed 10-times-bigger folds in comparison with normalization with EEF2/RPLP0.

To study the effects of selected reference genes on target gene expression and assess the appropriateness of the most stable combination of reference miRNAs (miR-191-5p/miR-103-3p/miR-16-5p), we examined the expression patterns of two target miRNAs—miR-181a-5p and miR-223-3p—which were reported to be upregulated in IBD [23,35]. We normalized the expression of target miRNAs in groups of samples that differ in factors that have been shown to significantly affect the expression stability of reference genes (i.e., DSS treatment and sex), to the most stable combination of reference miRNAs (miR-191-5p/miR-103a-3p/miR-16-5p) and to the least stable reference, U6 (Figure 4b). In comparison with normalization to U6, normalization to the most stable combination of reference miRNAs (miR-191-5p/miR-103-3p/miR-16-5p) did not show significant difference in the relative expression of miR-223-3p. However, we observed a difference in significance levels of relative expression of miR-181a-5p between the control and DSS group in females, when comparing the expression of miR-181a-5p normalized to the most stable combination or reference miRNAs (miR-191-5p/miR-103a-3p/miR-16-5p), or when comparing the expression of miR-181a-5p normalized to the least stable reference, U6. 

Our results demonstrate the importance of the selection of appropriate reference genes as they have a significant impact on the expression levels of target genes.

## 4. Discussion

This is the first study that evaluated the stability of reference genes (mRNA/miRNA) for their expression stability in archived FFPE colon samples involving 117 C57BL/6JOlaHsd mice (males and females), and evaluated the influence of inter/intra-experimental factors on expression levels of target genes. This was possible because archived FFPE samples were selected from nine DSS experiments with known inter/intra-experimental factors. Among 13 candidate reference genes evaluated in fresh colon samples [7], we selected reference genes based on the following criteria: 1. reference genes with the highest expression stability; 2. with the Cq-values lower than 30; and 3. with no/the lowest influence by the DSS treatment. Namely, the FFPE procedure causes chemical modification, fragmentation, and degradation of RNA, resulting in low yields of RNA [44,45] and 4-5-cycle-higher Cq-values in FFPE tissues compared to matched fresh frozen tissues [46,47]. Higher Cq-values of selected reference genes (i.e., EEF2, TBP, RPLP0, NONO, and PPIA) in archived FFPE tissues were also observed in our study (6-8 cycles higher) when results were compared with fresh tissues analyzed by Eissa et al. [7]. Since RNA in archived FFPE samples is degraded [16], the use of amplicons smaller than 80 base pairs is recommended. We used hydrolysis detection probes with lengths from 64-112 bp because, for some reference genes, probes shorter than 80 bp were not available (i.e., RPLP0-89bp and PPIA-112bp). 

Our study demonstrated that in archived FFPE samples ranking order of reference, gene expression stability was similar to those observed in fresh samples [7]. The most stable reference gene in our study was EEF2, irrespective of the method applied (NormFinder, BestKeeper), followed by RFLP0, TBP, PPIA, NONO (Normfinder) or TBP, NONO, RPLP0, PPIA (BestKepper). Similar stability ranking was demonstrated by Eissa [7] in fresh samples using GeNorm (EEF2, TBP, RPLP0, PPIA, NONO) or BestKeeper (EEF2, TBP, NONO, RPLP0, PPIA). In addition, EEF2 and TBP have been demonstrated to be stably expressed by others in various organs of C57BL/6 mice [48,49]. Nevertheless, the second most stable gene in fresh samples was TBP, which showed great potential in most of the archived FFPE samples in our study as well. It was the second most stable reference gene in females. However, relatively high Cq-values of TBP seen already in fresh samples resulted in undetermined results in some of the FFPE samples from males (not expressed in 16/117 samples), which made TBP unsuitable as a reference gene for archived FFPE samples. RPLP0 has been found to be the second most stable gene in archived FFPE samples (NormFinder), and the third most stable in fresh samples (geNorm). Interestingly, RPLP0 was proposed as a reference gene for colon tissue of IBD patients [10,50] without significant influence on the disease/inflammation [50]. Inflammation is known to affect the expression stability of reference genes in various organs in rodents and humans [7,8,10,50,51,52,53,54,55,56]. In fresh samples, the expression stability of EEF2 and TBP was not affected by DSS treatment, while RPLP0, NONO, and PPIA were significantly affected [7]. On the contrary, in our study, RPLP0 and TBP were not affected by DSS treatment, while EEF2, NONO, and PPIA were. It is important to note that sex, number, and phase of disease/colitis differed between both studies. In our study, 117 FFPE samples from females and males at various phases of inflammation (disease) were analyzed, while in fresh samples only 6 male mice, at one time point of inflammation, were analyzed [7]. A discrepancy in the results between the studies may thus be due to the sex of the animals involved, the phase of disease included, and the number of animals analyzed, and not the FFPE procedure. It was already shown that reference gene expression stability may differ between males and females in various organs [6,57,58,59] including the intestine [11], which is in agreement with our results. In archived FFPE colon samples, all candidate reference genes, including miRNA, showed more stable expression in females (Appendix A), while, in the case of U6, males showed more stable expression than females. In males, PPIA was not affected by DSS treatment, while in females it was. Finally, among 13 candidate reference genes, a combination of EEF2/TBP (geNorm) was proposed as a reference gene for fresh colon samples of DSS-treated male C57BL/6 mice [7], while for archived FFPE colon samples of DSS-treated C57BL/6 mice of both sexes, with various phases of the disease/colitis and inter/intra-experimental variability involved, the combination of EEF2/RPLP0 (NormFinder) was proposed as a robust reference gene. 

Due to its length (miRNA; 20-23 nt) and protein protection by the RISC complex [60,61], miRNA is less susceptible to RNA degradation in archived FFPE samples than mRNA. Four potential reference genes for miRNA normalization were selected: U6 as the most commonly used in the DSS mouse model [23,24,25,26,27], and miR-191-5p, miR-103a-3p, miR-16-5p as validated in human pediatric IBD or colorectal cancer [21,22].

To our knowledge, we are the first to evaluate reference miRNA/U6 expression stability for the DSS model. Our results show that U6 is the least suitable reference for the normalization of miRNA. U6 has been found inappropriate as a reference for the normalization of miRNA in humans (carcinoma tissue [62], serum [63]), and a mouse model of peripheral nerve injury [64]. It is important to note that U6 is a small nuclear RNA that differs from miRNAs in biochemical characteristics, the efficiency of extraction, and RT [65], all of which suggest that U6 may not be a suitable reference for miRNA normalization in general. Our results confirmed that, in archived FFPE samples, miRNA shows more stable expression than mRNA, which implies that archived FFPE samples represent an invaluable source for molecular research of miRNA without an additional use of animals. Our results show that sex did not affect the expression stability of reference miRNA, but only in U6. In our study, only miR-103a-3p was significantly altered by DSS treatment. Our results are thus in agreement with those obtained in humans, where miR-191-5p and miR-16-5p were among the most stable, but miR-103a-3p among the least stable miRNA in fresh frozen samples of pediatric IBD [22]. NormFinder and BestKeeper calculations proposed a combination of miR-191-5p/miR-103a-3p/miR-16-5p as the most suitable miRNA reference for normalization of qPCR results obtained from archived FFPE colon samples of DSS-treated C57BL/6JOlaHsd mice. 

To evaluate and assess the appropriateness of the identified best combination of reference genes for mRNA (EEF2/RPLP0) and miRNA (miR-191-5p/miR-103-3p/miR-16-5p), we used two target genes (TNFR1, TNFR2) and miRNAs (miR-181a-5p,miR-223-3p), which were reported to be upregulated in IBD [23,35], and examined their relative expression according to the usually applied method of normalization, and additionally with the use of multivariate analysis. The usually applied method of normalization demonstrates how the selection of reference gene affects the relative expression of target genes. However, this method does not tell us if the influence of the included inter/intra-experimental factors (which have an effect on the expression patterns of genes) were neutralized.

Among the inter/intra-experimental factors regarding the archived FFPE samples are variability in formalin fixation process [46,66], storage conditions, and storage duration of FFPE blocks [67]. However, we did not find any correlation between the expression pattern of reference genes and the storage duration of archived FFPE samples. Likewise, studies on archived FFPE samples with variations of formalin fixation time (ranging from 1–3 weeks) showed that the time of fixation did not affect the quality of the results but only the yield of RNA [68]. The long-term storage of FFPE blocks (from 1 to 26 years) had no significant effect on the quality and quantity of the extracted macromolecules also in other studies [15,68]. Storage duration of FFPE blocks did not affect miRNA expression patterns [61,69,70]. 

Interexperimental factors systematically included in our study are one of the most frequent differences in the DSS mouse model [18]. In addition, the advantage of using FFPE samples as a source of RNA is that FFPE samples enable direct evaluation of gene expression patterns in regard to the histological picture. It is important to note that samples taken for mRNA analyses (fresh frozen) do not necessarily represent the histological picture seen in the samples taken for histology (formalin fixation). Histology shows that focally distributed lesions have different intensities and locations, or can be even missed by the sampling. We must not forget that mice are small and most of the histologically diagnosed lesions cannot be seen by the naked eye. 

Our results showed that the expression pattern of all tested reference genes among the experiments differed, indicating that inter/intra-experimental factors influenced the expression stability of reference genes (mRNA/miRNA) (which may in part explain discrepancies among studies). To determine which inter/intra-experimental factors influence the expression patterns of the target genes in FFPE samples, multivariate analysis was performed. In addition, multivariate analysis enabled us to investigate whether normalization with selected reference genes neutralized the influence of particular inter/intra-experimental factors on target genes. To our knowledge, we are the first to investigate the effects of inter/intra-experimental factors on target gene expression. 

However, multivariate analysis showed that both absolute and relative expression levels of target genes were influenced by most of the inter/intra-experimental factors. Depending on the reference gene selection, inter/intra-experimental factors showed different patterns and levels of contribution to the expression level of the target gene. In our case, multivariate analysis showed that factors such as experiment (E1–E9), sex, and histological picture have the greatest influence on absolute levels of target genes if they are not normalized. Normalization decreased, and in some cases altered, the influence of factors on target gene expression, demonstrating that normalization successfully neutralized inter/intra-experimental differences, but only to a certain extent. At the same time, multivariate analysis also showed which inter/intra-experimental factors still had a certain level of influence on the relative expression of target genes. This is in accordance with the fact that genome expression is a highly integrated system, in which core genes can be influenced (even if at a lesser extent) by the motion of the others [71,72]. Our study thus demonstrates the importance and complexity of the reference gene selection process. Since the use of optimal/robust reference genes for normalization is the ground for reliable qPCR results, the results also indicate that more research and more intensive investigation focused on the identification of robust reference genes in rodent models needs to be performed in the future. As demonstrated, archived FFPA samples offer an invaluable opportunity for systematically oriented investigation of optimal/robust reference genes without the additional use of animals.

## 5. Conclusions

In conclusion, our study showed that archived FFPE samples are a reliable source of RNA for further research if the analyses are done in accordance with the latest knowledge and methodology. Results confirmed that the FFPE process affects Cq-values (6–8 cycles higher), which needs to be taken into consideration when working with archived FFPE samples or using data evaluated on fresh tissues. Our results imply that the FFPE procedure does not change the ranking of stability of reference genes obtained in fresh tissues, which suggests that FFPE samples can be used as a source for further reference gene identification process. Since miRNA is more stable than mRNA, archived FFPE samples offer an invaluable source for miRNA research (i.e., retrospective analysis, identification of biomarkers, evaluation of robust reference genes) without any additional use of animals. 

In addition, multivariate analysis showed that the histological picture is an important factor affecting the expression levels of target genes. Thus, FFPE samples as a source for RNA analyses have an added advantage not only in terms of reduction of the use of animals, but also in terms of accuracy of results (expression in correlation to the histological picture). Finally, multivariate analysis showed that normalization of qPCR data with currently proposed reference genes may still be influenced by some inter/intra-experimental factors, and therefore further investigation to find optimal/robust reference genes is highly recommended. To confirm our findings, more research on archived FFPE samples is needed.

## Figures and Tables

**Figure 3 biology-12-00190-f003:**
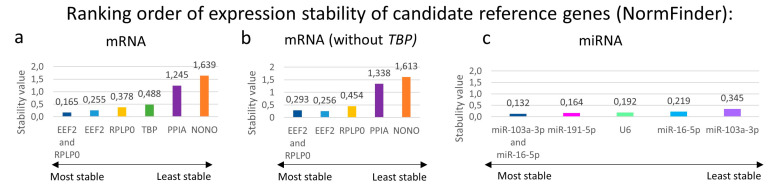
Expression stability of candidate reference genes in archived FFPE colon samples of control and DSS-treated C57BL6/JOlaHsd mice, calculated with NormFinder. The lower the stability values, the better the stability of reference genes. (**a**) Stability values of mRNA candidate reference genes and best combination of two mRNA reference genes in FFPE samples with determined TBP (n = 101) and (**b**) without TBP (n = 117). (**c**) Stability values of miRNA candidate reference genes and best combination of two miRNA reference genes in all FFPE samples (n = 117).

**Figure 4 biology-12-00190-f004:**
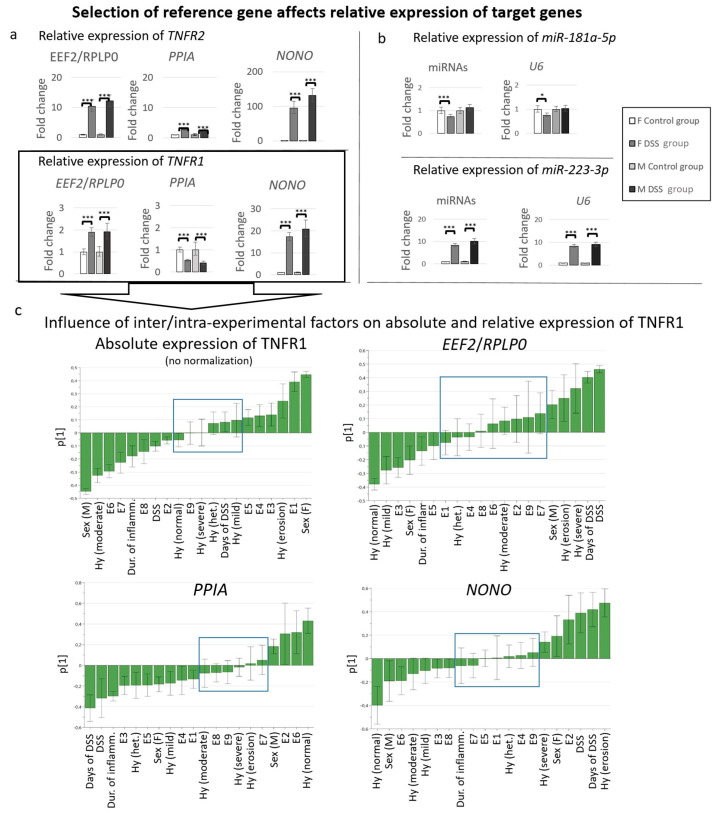
Selection of reference genes affects relative expression levels of target genes and the impact of inter/intra-experimental factors on target gene expression in archived FFPE samples of control and DSS-treated C57BL6/JOlaHsd mice (females and males). (**a, b**) Graphs present relative expression level of (**a**) TNFR1 and TNFR2, (**b**) miR-181a-5p and miR-223p (**a**) normalized to the most (EEF2/RPLP0) and to the least stable (PPIA and NONO) mRNA reference genes and (**b**) to the most (miR-191-5p/miR-103a-3p/miR-16-5p) and to the least stable (U6) miRNA reference genes, respectively. Different normalization led to significant differences. (**c**) Graphs represent influence of inter/intra-experimental factors on absolute (no normalization) and relative expression levels of TNFR1 (normalized to the combination of reference genes EEF2/RPLP0, and PPIA and NONO). Normalization of TNFR1 reduced the influence of inter/intra-experimental factors on relative expression levels of TNFR1 (factors inside Table 1. The number of inter/intra-experimental factors that do not affect TNFR1 is greater in the case of normalization with the most stable mRNA reference genes (EEF2/RPLP0). Legend: F, female; M, male; E1–E9, experiment number; DSS, DSS treatment applied; duration of inflammation, days/months of water after DSS treatment; Hy, histology; normal, no lesion or inflammation; mild, moderate and severe refer to inflammation; erosion, destruction of epithelial layer of mucosa; heterogeneous, includes both erosion and inflammation. Statistical significance: * *p* ≤ 0.05, ** *p* ≤ 0.01, *** *p* ≤ 0.001.

**Table 3 biology-12-00190-t003:** Basic statistics of the Cq-values of candidate reference genes in archived FFPE colon samples of control and DSS-treated C57BL6/JOlaHsd mice (n=117). Legend: SD, standard deviation; TBP+, 16 of 117 FFPE samples were undetermined for TBP, therefore, only 101 FFPE samples in which TBP was expressed were included in the calculation.

mRNA/miRNA	Minimum	Maximum	Median	Mean	SD
EEF2	24.02	32.34	26.28	26.94	1.748
TBP+	31.99	40.71	33.67	34.20	1.722
NONO	24.67	34.87	29.45	29.75	2.244
PPIA	24.43	34.58	27.07	27.31	2.324
RPLP0	23.76	32.92	26.48	26.85	2.409
miR-191-5p	27.31	32.29	28.46	28.56	0.849
miR-103a-3p	26.86	33.27	28.78	28.90	1.017
miR-16-5p	25.39	30.95	26.37	26.54	0.813
U6	24.79	32.43	27.08	27.18	1.238

**Table 4 biology-12-00190-t004:** Expression stability of candidate reference genes in archived FFPE colon samples of control and DSS-treated C57BL6/JOlaHsd mice, calculated with BestKeeper. Legend: F, female; M, male; CP, crossing point (BestKeeper’s notation for Cq); SD, standard deviation; grey cells denote control groups; white cells denote DSS groups; red cells denote low expression stability (high SD-values > 1); TBP+, number of samples (n) included in the calculation of TBP is smaller than shown in the first column (16 of 117 FFPE samples were undetermined for TBP, 2 from the control and 14 from DSS group, all undetermined FFPE samples were males).

BestKeeper		EEF2	TBP+	NONO	PPIA	RPLP0	miR-191-5p	miR-103a-3p	miR-16-5p	U6
Control group (n = 20)	Geo Mean [CP]	26.93	34.04	30.29	30.59	28.90	28.44	28.11	26.52	26.92
	SD [± CP]	1.22	1.16	1.09	1.81	1.69	0.57	0.59	0.50	0.58
DSS group (n = 97)	Geo Mean [CP]	27.08	34.19	30.39	30.41	28.70	28.58	29.05	26.53	28.01
	SD [± CP]	1.37	1.38	1.45	2.09	2.17	0.70	0.75	0.63	1.04
F control group (n = 10)	Geo Mean [CP]	25.81	33.03	29.38	29.05	27.36	28.35	28.00	26.27	26.77
	SD [± CP]	0.69	0.59	0.91	1.34	1.20	0.45	0.42	0.41	0.43
F DSS group (n = 47)	Geo Mean [CP]	25.94	33.22	29.34	28.83	26.94	28.11	28.60	26.04	27.58
	SD [± CP]	0.52	0.60	0.76	1.00	0.82	0.46	0.66	0.35	1.25
M control group (n = 10)	Geo Mean [CP]	28.10	35.34	31.22	32.22	30.53	28.52	28.23	26.76	27.08
	SD [± CP]	1.70	0.73	1.48	2.56	2.29	0.71	0.76	0.63	0.69
M DSS group (n = 50)	Geo Mean [CP]	28.20	35.50	31.41	31.97	30.45	29.02	29.48	27.00	28.43
	SD [± CP]	1.33	1.42	1.35	1.98	2.01	0.63	0.65	0.57	0.59

## Data Availability

The datasets used and analyzed during the current study are available in Appendix A.

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
