# Peer review of "Selection and Evaluation of mRNA and miRNA Reference Genes for Expression Studies (qPCR) in Archived Formalin-Fixed and Paraffin-Embedded (FFPE) Colon Samples of DSS-Induced Colitis Mouse Model"

_biology, 2023, doi:10.3390/biology12020190_

Round 1
Reviewer 1 Report
The manuscript by Ana Unkovič and colleagues evaluated the mRNA and miRNA reference genes for qPCR studies in archived formalin-fixed and paraffin-embedded (FFPE) colon samples of the DSS-induced colitis mouse model. The data appeared to show that the FFPE procedure does not change the ranking of stability of reference genes obtained in fresh tissues, and therefore the authors concluded that FFPE samples as a source for RNA analyses have an added advantage not only in terms of reduction of animals but also in terms of accuracy of results. Overall, the results presented here are interesting, and they might be a good contribution to the field and bring some new information on relevant studies. Below please find my comments:
The authors compared the characteristics of mRNA and miRNA reference gene expressions using archived formalin-fixed and paraffin-embedded (FFPE) colon samples obtained from the DSS-induced colitis mouse model, but make a universal conclusion that FFPE samples as a source for RNA analyses have an additional advantage in the Abstract section (Lines 41-42). Although we generally thought the FFPE samples might have some advantages over formalin-treated samples for omics studies, is it way too exaggerated to make such a conclusion given that the authors conducted the studies using specific tissue samples along with a specific disease model?
Line 115: spelling mistake for “authopsy”.
Author Response
The authors compared the characteristics of mRNA and miRNA reference gene expressions using archived formalin-fixed and paraffin-embedded (FFPE) colon samples obtained from the DSS-induced colitis mouse model, but make a universal conclusion that FFPE samples as a source for RNA analyses have an additional advantage in the Abstract section (Lines 41-42). Although we generally thought the FFPE samples might have some advantages over formalin-treated samples for omics studies, is it way too exaggerated to make such a conclusion given that the authors conducted the studies using specific tissue samples along with a specific disease model?
We appreciate your comment. We deleted the last sentence.
Line 115: spelling mistake for “authopsy”.
Thank you. The typo has been corrected.
Reviewer 2 Report
Authors present interesting evidences supporting the idea that the election of a good reference gene in qPCR studies is not easy. In this work, the authors present solid results supporting the hypothesis that the histological treatment of a tissue sample can affect the expression level of housekeeping genes leading to wrong interpretation of qPCR results. For example, the results presented in figure 4 are the best evidence that it is important the evaluation of reference genes when the study is performed using fixed tissues or any other histological preparation.
I only have a few suggestions in order to improved the manuscript:
Main Issue:
1. The work is based in the analysis of RNA extracted from FFPE samples, however, the quality of the RNA is not showed as a result. I think this point is very important because the FFPE treatment could affect the quality of the RNA leading to problems in the amplification of the RNA in the RT-qPCR. Maybe this analysis could be showed in figure 1.
Minor Issues:
1. In Figure 1b, what is the X axis?. It is necessary the addition of a description of the X axis in the legend of the figure because is not clear for me what is the variable indicated in the axis and it is difficult to understand the result indicated in the graphics of figure 1b.
2. In figure 2, the color of the graphic bars must be equal than panels "a" and "e". Are there any issue associated to the choice of this color pattern in the figure that explain this?
3. One of the graphics in Figure 4c does not have the title. I think must be PPIA.
4. The order of appearance of the Figures in the text is not appropriated. For example, Figure 2 is mentioned in lane 335 and Figure 1 appears in lane 400 and Figure 2 in lane 408. Remember always relate the text with the figures to better understand the results of the work.
Author Response
Authors present interesting evidences supporting the idea that the election of a good reference gene in qPCR studies is not easy. In this work, the authors present solid results supporting the hypothesis that the histological treatment of a tissue sample can affect the expression level of housekeeping genes leading to wrong interpretation of qPCR results. For example, the results presented in figure 4 are the best evidence that it is important the evaluation of reference genes when the study is performed using fixed tissues or any other histological preparation.
I only have a few suggestions in order to improved the manuscript:
Main Issue:
- The work is based in the analysis of RNA extracted from FFPE samples, however, the quality of the RNA is not showed as a result. I think this point is very important because the FFPE treatment could affect the quality of the RNA leading to problems in the amplification of the RNA in the RT-qPCR. Maybe this analysis could be showed in figure 1.
Thank you for your suggestion. We totally agree that the quality of RNA, particularly in FFPE samples, is of great importance. However, there are different methods used for the assessment of RNA quality in FFPE samples. We used indirect methods of RNA quality assessment. First, we assessed the purity of extracted RNA from FFPE samples spectrophotometrically by measuring absorbance at 230 and 280 nm using NanoDrop One (Thermo Fisher Scientific, Waltham, Massachusetts, USA). Since all our FFPE samples had an absorption ratio of A260/280 1.8 or higher, results indicate that isolated RNA from FFPE samples was of good purity. The integrity of isolated RNA can be assessed using differential amplicons (ΔAmp), i.e. a molecular approach used for the assessment of the quality of targeted RNA species (https://pubmed.ncbi.nlm.nih.gov/27077042/). We used a similar approach and calculated the difference in Cq-values between two amplicons of different sizes (74 and 112 bp) and observed stable difference among samples. Although amplicons were not from the same target (EEF2 and PPIA, respectively) these calculations indicate that RNA is of good integrity. Additionally, we used PPIA as our control (as it had the longest amplicon – 112 bp). Since PPIA was successfully amplified in all samples, our results indirectly indicate that the tested RNA was of good quality. All Cq-values for calculations are available in the Supplementary material.
Unfortunately, we do not have an idea how to show these results in Figure1, thus we added only an additional description of the assessment of RNA integrity in the manuscript, i.e. in the Methods section and Results (text in red color).
Minor Issues:
- In Figure 1b, what is the X axis?. It is necessary the addition of a description of the X axis in the legend of the figure because is not clear for me what is the variable indicated in the axis and it is difficult to understand the result indicated in the graphics of figure 1b.
Thank you for your comment. A description of the x-axis is now added in the text of Figure 1b.
- In figure 2, the color of the graphic bars must be equal than panels "a" and "e". Are there any issue associated to the choice of this color pattern in the figure that explain this?
The color palette of Figure 2a and 2e is in agreement with the color palette of Figure 1b. Figure 2a and 2e present both, samples from the control and DSS group. Figures 2b-d and 2f-h are deliberately not colored as we wanted to emphasize the differences between the control and the DSS groups.
- One of the graphics in Figure 4c does not have the title. I think must be PPIA.
Thank you. The missing title (PPIA) has been added to Figure 4c.
- The order of appearance of the Figures in the text is not appropriated. For example, Figure 2 is mentioned in lane 335 and Figure 1 appears in lane 400 and Figure 2 in lane 408. Remember always relate the text with the figures to better understand the results of the work.
Thank you. We corrected and added a, b letters for better understanding and highlighted the text where Figure 1 appears.
